# Bulk and molecular-level composition of primary organic aerosol from wood, straw, cow dung, and plastic burning

Jun Zhang[1], Kun Li[1,a], Tiantian Wang[1], Erlend Gammelsæter[1,b], Rico K. Y. Cheung[1], Mihnea Surdu[1], Sophie Bogler[1], Deepika Bhattu[2], Dongyu S. Wang[1], Tianqu Cui[1], Lu Qi[1], Houssni Lamkaddam[1], Imad El Haddad[1], Jay G. Slowik[1], Andre S. H. Prevot[1], David M. Bell[1]

[1]Laboratory of Atmospheric Chemistry, Paul Scherrer Institute, Villigen, 5232, Switzerland

[2]Department of Civil and Infrastructure Engineering, Indian Institute of Technology Jodhpur, 342037, India

[a]now at: Environmental Research Institute, Shandong university, Qingdao, 266237, China

[b]now at: Department of Chemistry, Norwegian University of Science and Technology, Trondheim, 7491, Norway

*Correspondence to*: Andre S. H. Prevot (andre.prevot@psi.ch) and David M. Bell (david.bell@psi.ch )

**Abstract.** During the past decades, the source apportionment of organic aerosol (OA) in the ambient air has been improving substantially. The database of source retrieval model resolved mass spectral profiles for different sources has been built with the aerosol mass spectrometer (AMS). However, distinguishing similar sources (such as wildfires and residential wood burning) remains challenging, as the hard ionization of AMS mostly fragments compounds and therefore cannot capture the detailed molecular information. Recent mass spectrometer technologies of soft ionization and high mass resolution have allowed for aerosol characterization at the molecular formula level. In this study, we systematically estimated the emission factors and characterized the primary OA (POA) chemical composition with the AMS and the extractive electrospray ionization time-of-flight mass spectrometer (EESI-TOF) for the first time from a variety of solid fuels, including beech logs, spruce and pine logs, spruce and pine branches and needles, straw, cow dung, and plastic bags. The emission factors of organic matter estimated by AMS and hydrocarbon gases estimated by the total hydrocarbon analyzer are $16.2 \pm 10.8$ g kg$^{-1}$ and $30.3 \pm 8.5$ g kg$^{-1}$ for cow dung burning, which is generally higher than that of wood (beech, spruce, and pine), straw, and plastic bags burning (in the range from 1.1 to 6.2 g kg$^{-1}$ and 14.1 to 19.3 g kg$^{-1}$). The POA measured by the AMS shows that the $f_{60}$ (mass fraction of $m/z$ 60) varies from 0.003 to 0.04 based on fuel types and combustion efficiency for wood (beech, spruce, and pine) and cow dung burning. On molecular level, the dominant compound of POA from wood, straw, and cow dung is $C_6H_{10}O_5$ (mainly levoglucosan), contributing ~7% to ~30% of the total intensity, followed by $C_8H_{12}O_6$ with fractions of ~2% to ~9%. However, as they are prevalent in all burns of biomass material, they cannot act as tracers for the specific sources. By using the Mann-Whitney U test among the studied fuels, we find specific potential new markers for these fuels from the measurement of the AMS and EESI-TOF. Markers from spruce and pine burning are likely related to resin acids (e.g. compounds with $20 - 21$ carbon atoms). The product from pyrolysis of hardwood lignins is found especially in beech logs burning. Nitrogen-containing species are selected markers primarily for cow dung open burning. These markers in the future will provide support for the source apportionment.

**Key words**: AMS, EESI-TOF, biomass burning, source apportionment, markers

## 1 Introduction

Emissions from combustion are a major source of primary organic aerosol (POA), black carbon (BC), inorganic aerosol, and inorganic / organic gases (Andreae, 2019; Bond et al., 2007). After being emitted to the atmosphere, volatile organic compounds (VOCs) can further react to form lower volatility components and generate secondary organic aerosol (SOA). The primary emissions and their subsequent transformations have significant implications for air quality, climate, and human health (Chen et al., 2017). Accordingly, a large number of laboratory and field measurements have been carried out to disentangle the roles of burning-induced aerosols in polluted areas.

Solid fuel combustion is a major source of air pollution in many places over the world. In Southeast Asia, haze events are mainly attributed to the wildfires, agricultural waste burning, and peatland fires (Adam et al., 2021). In India, more than half of households use inefficient stoves for cooking, burning solid fuels such as firewood, charcoal, crop residues, and cow dung (Census of India, 2011). This contributes to poor household air quality, chronic and acute respiratory diseases, and even premature death (Smith et al., 2014). Plastic burning has been estimated to contribute 13.4% of fine particulate matter ($PM_{2.5}$) yearly in India, 6.8% in wintertime in China (Haque et al., 2019), and 2% to 7% in wintertime in the US (Islam et al., 2022). The toxic pollutants released from plastic burning, including olefins, paraffin, and polycyclic aromatic hydrocarbons, can cause respiratory irritation, and carcinogenic and mutagenic effects (Pathak et al., 2023). The extent to which primary particulate matter adversely affects health is source-dependent. Recent studies have shown that biomass burning-related particles have been linked to reactive oxygen species and oxidative stress, increasing the risks of cardiovascular diseases (Daellenbach et al., 2020; De Oliveira Alves et al., 2017; Tuet et al., 2019). Therefore, identifying the sources of aerosols is essential for assessing health risks and developing mitigation strategies.

Organic aerosol (OA) source apportionment has been widely studied using receptor models, e.g. positive matrix factorization (PMF), with OA composition characterized by an aerosol mass spectrometer (AMS) or aerosol chemical speciation monitor (ACSM). Many studies have successfully resolved source-related factors, for example, hydrocarbon-like OA (HOA), oxygenated OA (OOA), biomass burning OA (BBOA), coal combustion OA (CCOA), and so on, via PMF (Chen et al., 2021; Huang et al., 2014; Ng et al., 2010; Tobler et al., 2020; Wang et al., 2019; Crippa et al., 2014). The identification and validation of resolved factors rely strongly upon the spectral characteristics of source emissions. For example, hydrocarbon ion series $C_nH_{2n+1}^+$ and $C_nH_{2n-1}^+$, e.g. $C_4H_9^+$ at $m/z$ 57 and $C_3H_5^+$ at $m/z$ 41, are often referenced as tracers for HOA (Mohr et al., 2012), while $C_2H_4O_2^+$ at $m/z$ 60 is the main marker for wood and other biomass burning, as $C_2H_4O_2^+$ is a characteristic major fragment of anhydrosugars (e.g. levoglucosan) produced from cellulose pyrolysis (Alfarra et al., 2007). However, achieving finer separation and interpretation of sources within one of the OA categories mentioned above from highly mixed aerosols in the environment remains challenging, because the laboratory mass spectral profile database of primary emissions is limited and the potential molecular specificity is impeded by intensive fragmentation in the AMS and ACSM.

To minimize the loss of the molecular information from fragmentation, soft ionization and novel sampling techniques have been deployed to measure the chemical composition of particles in greater detail. A thermal desorption aerosol gas chromatograph (TAG) coupled to a AMS has been used and provided the molecular characterization of OA and SOA (Bertrand et al., 2018). The filter inlet for gases and aerosols (FIGAERO) measures molecular composition of OA via thermal desorption coupled to a chemical-ionization mass spectrometer (Lopez-Hilfiker et al., 2014). Nonetheless, thermal decomposition can occur during the thermal

desorption process (Stark et al., 2017), causing potential artifacts and hindering the identification of components.
Liquid chromatography mass-spectrometer can avoid thermal desorption and separate mixtures including isomers
based their chemical affinity with the mobile and stationary phases (Zhang et al., 2021). However, it requires pre-
treatment of samples which could introduce artefacts and lowers the time resolution. An extractive electrospray
ionization time-of-flight mass spectrometer (EESI-TOF) has been recently developed for online OA measurement
with generally insignificant decomposition or fragmentation (Lopez-Hilfiker et al., 2019). As a result, it provides
a molecular-level (i.e., molecular formula determination) mass spectrum with a time resolution of seconds.
Consequently, improved real-time investigations of chemical composition in chamber experiments (Surdu et al.,
2023; Bell et al., 2022) and SOA source apportionment in the field measurement (Tong et al., 2021; Kumar et al.,
2022; Qi et al., 2022) became possible. Thus far, a detailed study of primary emissions from complex sources, e.g.
combustion, has not yet been conducted with the EESI-TOF, which necessitates the measurement to fully utilize
the chemical resolution capabilities of EESI-TOF for characterizing mass spectra and supporting the source
apportionment in the field.
In this work, we systematically characterize the POA composition using both AMS and EESI-TOF from a variety
of burning fuels from both residential stoves (beech logs and a mixture of spruce and pine logs) and open
combustion (spruce and pine branches and needles, straw, cow dung, and polyethylene plastic bags). The emission
factors of trace gases are presented and possible molecular markers for the burning fuels in this study are discussed.
This work allows for a better understanding of the POA chemical composition emitted from different burning
sources, provides important reference spectra for source apportionment, and potential markers to use to assess the
importance of different biomass burning sources.
**2 Materials and methods**
**2.1 Experimental setup and instrumentation**
A total of 36 burning experiments were conducted using 6 different types of burning materials, including beech,
spruce, pine, straw, cow dung, and plastic bags. Beech logs, spruce and pine logs, fresh spruce and pine branches
and needles, as well as straw were sourced from a local forestry company in Würenlingen, Switzerland. Cow dung
cakes (made of cow dung and straw) were collected from Goyla dairy, Delhi, India, and polyethylene plastic bags
were bought in Delhi, India. To represent residential burning, logs of (1) beech and (2) spruce and pine were
burned separately in a stove (Bruns et al., 2017). Agricultural waste combustion and forest fires were respectively
represented by burning (1) straw and (2) a mixture of fresh branches and needles of spruce and pine in an open
stainless steel cylinder (65 cm in diameter and 35 cm in height). Finally, the half-open stove (e.g. angithi) and
waste burning in India and some other areas (Fleming et al., 2018), respectively, were represented by burning (1)
cow dung cakes and (2) plastic bags on top of the stainless steel cylinder, with the fuel resting on a mesh steel
plate. The experimental setup is shown in Figure S1. The fuels were ignited with fire starters / kindling and the
emissions were pulled into either a chimney (for stove burning) or a hood (for open burning). After starters /
kindling burnt away (~3 to 10 min. after ignition), the emissions were introduced into a holding tank through
stainless steel sampling lines heated to 180 °C and passing through an ejection dilutor (DI-1000, Dekati Ltd.) with
a dilution ratio of ~10. The holding tank is a stainless steel container (1 m$^3$) used to store emissions. It is also
designed for averaging the emissions at different combustion efficiency in order to fully characterize the emission
in the real ambient. The emissions were injected into the holding tank for 10 to 30 min, depending on the emission
source. Typically, the injection was stopped when the measured POA concentration was above ~20 μg/m³ after
~60 times dilution in the sampling lines. In different burning experiments, POA concentrations in the holding tank
varied between 1 to 5 mg m⁻³ prior to sampling line dilution. The holding tank was flushed overnight with clean
air before each experiment, ensuring the background particle concentrations were less than 10 #/cm³.
The emissions were delivered from the holding tank to sampling instruments via stainless steel lines (6 mm O.D.)
for particles and via Teflon lines (6 mm O.D.) for gases. Gas analyzers were used for monitoring the concentration
of CO (Horiba APMA-370), $CO_2$ (LI-COR LI-7000), and total hydrocarbon (THC, including methane) (Horiba
APHA-370). Particle concentrations were measured using a scanning mobility particle sizer (SMPS, model 3938,
TSI) scanning a range of 16 to 638 nm. An aethalometer (AE 33, Magee Scientific) was used to retrieve the
concentration of equivalent BC (eBC). A long time-of-flight aerosol mass spectrometer (LTOF-AMS, Aerodyne
Research, Inc.) with a mass resolution of ~5000 over the range of $m/z$ 100 to $m/z$ 450 was deployed for online,
non-refractory particle characterization and a subset of experiments were performed with high-resolution time-
of-flight AMS (HTOF-AMS, Aerodyne Research, Inc.) with a mass resolution of ~2000 over the range of $m/z$ 100
to $m/z$ 450. The aerosols sampled by both the SMPS and AMS were dried with a Nafion dryer (Perma Pure). The
aerosol was continuously sampled by the AMS through a 100 μm critical orifice and focused by $PM_1$ aerodynamic
lens. Therefore, the class of the PM in this study belongs to $PM_1$. Mass spectra of positive ion fragments ($m/z$ 10
to 450) were obtained with a TOF mass spectrometer and were analyzed with the software SQUIRREL
(SeQUential Igor data RetRiEvaL) v.1.63 and PIKA (Peak Integration by Key Analysis) v.1.23 for the IGOR Pro
software package (Wavemetrics, Inc.). A detailed description of AMS can be found in Decarlo et al. (2006).
EESI-TOF was deployed for a real-time and molecular-level (i.e., molecular formula) measurement of OA with
minimal analyte fragmentation or decomposition (Lopez-Hilfiker et al., 2019). Before entering the EESI-TOF,
the aerosol passes through an activated charcoal denuder to remove gas-phase species. The aerosol intersects a
spray of charged droplets generated by an electrospray probe. Particles coagulate with the electrospray (ES)
droplets, and water-soluble compounds are extracted into the solvent and then ionized via the Coulomb explosion
mechanism as the droplets evaporate. 100 ppm sodium iodide (NaI) in pure water (MilliQ) was used as the
electrospray solution, resulting in the formation of $[M + Na]^+$ (M is the analyte) adduct in the positive ionization
mode. The EESI-TOF mass analyzer achieved a mass resolution of ~10000 at $m/z$ 173 and 11000 at $m/z$ 323. The
EESI-TOF operated with a time resolution of 1 s, and alternated 1.5 min of background measurements (in which
the sampled air passes through a high efficiency particulate air (HEPA) filter to remove particles) with 3.5 min of
direct sampling. These data were pre-averaged to 5 s for further analysis. Ions are only considered as signals from
emissions when their intensity difference between the particle measurement and the corresponding background
measurement periods were 1.9 times bigger than the propagated standard errors over the measurement cycle. For
those selected ions, their mass flux to the detector was calculated as Equation 1:

$$Mass_x = \frac{I_x \times MW_x \times 10^{18}}{N_a} \qquad \textit{Equation (1)}$$


where $Mass_x$ and $I_x$ are respectively the mass flux (attograms per second, ag s⁻¹) and the ion flux of (counts per
second, cps) of a group of detected ions with the same molecular weight. $MW_x$ is the molecular weight of $x$ (with
the mass of the charge carrier, typically Na⁺, removed). $N_a$ is Avogadro's number. To assist with the peak
identification, filters were collected from emissions and were analyzed with an ultrahigh-resolution mass
spectrometer (Orbitrap). The Orbitrap (Orbitrap Exploris 120, Thermo Fischer) has a mass resolving power of
140000 at *m/z* 200, and was operated in positive mode scanning from *m/z* 50 to 450.

## 2.2 Data analysis

The emission factor (EF) of species *i* was calculated using the carbon mass balance method (Radke and Ward,
1993), expressed in the unit of g kg $^{-1}$, as shown in Equation 2.

$$EF_i = \frac{m_i \cdot W_C}{\Delta CO + \Delta CO_2 + \Delta THC + \Delta OC + \Delta BC}$$                  *Equation (2)*

where $\Delta CO$, $\Delta CO_2$, $\Delta THC$, $\Delta OC$, and $\Delta BC$ are the background-corrected carbon mass concentrations of CO,
$CO_2$, THC, OC (organic carbon), and BC. OC was calculated from the ratio of organic aerosol and the ratio of
organic mass (OM) to OC (OM/OC) measured by AMS (Canagaratna et al., 2015). $m_i$ is the mass concentration
of species *i*. $W_C$ is the carbon mass fraction of the burning fuel. The $W_C$ was reported 0.46 for wood (Bertrand et
al., 2017), 0.45 for straw (Li et al., 2007), 0.45 for cow dung (Font-Palma, 2019), and 0.84 for plastic bags (Li et
al., 2001). In experiments, where BC is not available, the sum of OC and BC is considered equal to the particulate
matter (PM) determined by SMPS. The effective density of particles applied in the SMPS is determined by
comparing mass and volume distributions from the AMS and SMPS (Bahreini et al., 2005). The densities could
be underestimated because of the non-spherical shape of particles, especially particles from plastic bags burning
mainly due to the high contribution of BC. As the contribution of particles to the total carbon is much smaller than
the gases, these two methods have little differences calculating the denominator in Equation 2. Therefore, the EFs
of CO, $CO_2$, and THC are consistent using both methods. However, it could be important for calculating the EFs
for particulate species because of the possible discrepancy between the mass measured by the SMPS and AMS
arising from, for example, the particle size and effective density. Additionally, the OM/OC acquired by the AMS
also would add uncertainty when converting OM to OC because the high range of *m/z* without peak fitting is not
included in OM/OC. More comparison is discussed in Sect. 3.1.
The combustion condition was characterized by the modified combustion efficiency (MCE, Equation 3) (Ward
and Hardy, 1991). When the MCE is higher than 0.9, the combustion is considered as predominantly flaming.
When the MCE is lower than 0.85, it is dominated by smoldering.

$$MCE = \frac{\Delta CO_2}{\Delta CO + \Delta CO_2}$$                  *Equation (3)*

## 2.3 Identification of potential markers

The identification of potential markers for emissions was performed by Mann-Whitney U test (Mann and Whitney,
1947; Wilcoxon, 1945) which has been applied in many fields and for the current study has the advantage that it
does not require a large volume of normally distributed samples (Rugiel et al., 2021; Tai et al., 2022). It tests the
null hypothesis that the two population medians are equal against the alternative hypothesis that the two
populations are not equal. When the *p*-value is smaller than the significance level of 0.1, the median of the tested
sample is significantly high or low in the two-tailed test. Ions from a class of fuel that satisfy the pairwise
comparison test between one fuel *j* and other types of fuels were considered to be significantly high-fraction or
low-fraction ions in the fuel *j* and therefore have the potential as markers for the fuel *j*. The fold change (FC) of
ion $i$ in the fuel $j$ was calculated as the Equation 4, where the $f_{i,j}$ is the fraction of ion $i$ in the mass spectra profiles
of the fuel $j$, and $f_{i,other}$ is the average fraction of ion $i$ in the mass spectra from the other fuels.

$$FC_{i,j} = \frac{f_{i,j}}{f_{i,other}} \qquad\qquad \textit{Equation (4)}$$

## 3 Results and discussion

### 3.1 Emission factors from combustion

The average EFs of CO, $CO_2$, THC, PM, OM, and eBC, as well as the MCE values of the 6 types of burning are
shown in Table 1, and the EFs and MCE values for each experiment are presented in Table S1.
The average MCE values depend on fuel types, with the lowest values of 0.87 ± 0.03 (average ± 1 σ) observed
from cow dung open burning and the highest values of 0.98 ± 0.02 from plastic bags open burning, consistent
with smoldering combustion for cow dung and flaming/melting processes for plastic bags. Accordingly, cow dung
had the highest average CO EF (92.3 ± 27.4 g kg$^{-1}$) and the lowest $CO_2$ EF (1366.2 ± 88.4 g kg$^{-1}$), and vice versa
for plastic bags. The strong relationship between the MCE and some EFs is also true for the THC. In general,
lower the MCEs correspond to higher THC EFs within a given class of burning fuel. Taking straw burning as an
example, as shown in Table S1, the EFs of THC vary from 0.7 to 39.3 g kg$^{-1}$, with the MCE varying from nearly
1.00 to 0.89 correspondingly, resulting the high standard deviation of the EFs. These EFs of gases are comparable
with the reported EFs from the literature under similar conditions (Hennigan et al., 2011; Fang et al., 2017;
Bertrand et al., 2017).
The average EFs of PM is in the range of 3.1 to 16.6 g kg$^{-1}$. In general, the PM emitted from cow dung is dominated
by OM, and the eBC is minor. For beech logs and straw, the OM EF is around 3 times higher than the eBC EF.
Noticeably, the emission of PM from plastic bags is not very high (2.7 g kg$^{-1}$), but the EF of OM and eBC is
similar (1.1 g kg$^{-1}$ v.s. 1.0 g kg$^{-1}$). Note that when eBC data is not available, the sum of OC and BC in the
denominator in Equation 2 is assumed to be equal to the PM measured by the SMPS. Table S1 lists the comparison
of EFs for particulate species where possible. For the experiments of cow dung open burning and plastic bags
open burning, the EFs are consistent using both methods with the difference of PM EF < 6%, and on average less
than 15% for OM EF. However, for some beech logs stove burning experiments (BS3 and BS4), the effective
density required in the calculation is not available, and the average density of other beech logs in this study is
used. This results in some variance between these two methods. In general, the EFs of PM, OM, and BC agree
well with some previous literature (Fang et al., 2017; Goetz et al., 2018; Tissari et al., 2008). Nevertheless, the
reported EF values are highly dependent on the burning method (e.g. stove type) and combustion efficiency
(Bertrand et al., 2017). Additionally, the reported EFs for plastics vary substantially with their composition, and
the EF of the pure PE plastic bags are not often reported (Jayarathne et al., 2018; Wu et al., 2021; Hoffer et al.,

215    2020).

### 3.2 Chemical composition of POAs from combustion

### 3.2.1 Chemical composition of POAs measured with the AMS

The chemical composition of POAs of burning emissions is characterized with the AMS and EESI-TOF
simultaneously. As shown in Figure 1, the average mass spectra from $m/z$ 10 to 120 measured with AMS is

grouped into $C_xH_y$, $C_xH_yO_1$, $C_xH_yO_{2+}$, $C_xH_yN_z$, and $C_xH_yO_{1+}N_z$ families based on their elemental composition. In all the POAs, the $C_xH_y$-family is the most abundant group, mainly from ions at m/z 29, 39, 41, 43, 55, 57, 67, and 69 originating primarily from hydrocarbon compounds, with the biggest contribution from plastic bags (92%), followed by cow dung (70%) and straw (61%), generally higher than that of wood (beech, spruce, and pine, 48% to 54%) burning. The $C_xH_yO_{2+}$ and $C_xH_yO_1$ families are the second largest compositions, with major ions at $m/z$ 28, 29, 43, 44, and 60, which are higher in wood and straw emissions compared to cow dung. Among these ions, the mass fractions of $m/z$ 44 ($f_{44}$, mostly $CO_2^+$), mass fraction of $m/z$ 43 ($f_{43}$, mostly $C_2H_3O^+$ and $C_3H_7^+$), and mass fraction of $m/z$ 60 ($f_{60}$, mostly $C_2H_4O_2^+$) have the largest impact on the oxidation state of the aerosol. The fragment $C_2H_4O_2^+$ at $m/z$ 60 is widely used as a levoglucosan related marker for biomass burning and is most prominent in the wood burning emissions compared to the other burning fuels. Figure 2a shows that the POAs in this study are at the left bottom of the ambient OOA range (Ng et al., 2010) in the $f_{44}$ vs $f_{43}$ plot, indicating the POA is less oxygenated, which is consistent with previous studies (Hennigan et al., 2011; Fang et al., 2017; Xu et al., 2020). As shown in Figure 2b, the $f_{60}$ for the biomass source studied is greater than the background level (Cubison et al., 2011), suggesting the $f_{60}$ filter ($f_{60} = 0.003$) in the ambient is unlikely to miss biomass combustion. The contribution of $f_{60}$ relates to the burning fuel types and the combustion efficiency. For example, the $f_{60}$ from wood burning ranges from 0.02 to 0.04, generally higher than that of cow dung. The $f_{60}$ of straw open burning is distributed from below 0.01 to 0.025, resulting from the low to high MCE values correspondingly. The pie charts in Figure 1 indicate that the N-containing fragments from AMS are mainly from $C_xH_yN_z$ and $C_xH_yO_{1+}N_z$ family and they are the largest in the emission of cow dung open burning (4.3%) and followed by straw (3.4%), while the wood is relatively minimal ($\leq$ 1%) (Stockwell et al., 2016). The nitrogen in organonitrates would appear mainly at fragments of $NO^+$ and $NO_2^+$. However, the $NO^+$ and $NO_2^+$ originating from organonitrates are estimated almost 20 times smaller than $C_xH_yN_z$ family in cow dung burning using the $NO^+$ and $NO_2^+$ ratio between inorganic nitrates and organonitrates (Farmer et al., 2010), suggesting their contributions are minor to organic nitrogen.

In the region from $m/z$ 120 to 450 as shown in Figure S3, polycyclic aromatic hydrocarbons (PAHs) are observed. Based on the spectra of laboratory standards (Dzepina et al., 2007; Aiken et al., 2007), parent ions at $m/z$ 226, 252, 276, 300, and 326 correspond respectively to $C_{18}H_{10}$ (benzo[ghi]fluoranthene and cyclopenta[cd]pyrene) $C_{20}H_{12}$ (benzofluoranthene and benzopyrene), $C_{22}H_{12}$ (indenopyrene and benzoperylene), $C_{24}H_{12}$ (coronene), and $C_{26}H_{14}$ (dibenzoperylene). The fragment of m/z 239 could be methylbenzo[ghi]fluoranthene ($C_{19}H_{12}$) (Dzepina et al., 2007; Ji et al., 2010) or a fragment of dehydroabietic acid which has been found in fresh pine resin (Colombini et al., 2005). The m/z 219 and m/z 285 also could arise from the fragmentation of retene and dehydroabietic acid, respectively, which also can be derived from conifer resin (Dzepina et al., 2007; Jen et al., 2019; Zetra et al., 2016). These ions contribute 0.69% ± 0.14% and 0.66% ± 0.11% respectively to the total POA of the spruce and pine branches and needles open burning as well as spruce and pine logs stove burning, which is distinct from straw (0.36% ± 0.13%), beech logs (0.34% ± 0.14%), and cow dung (0.25% ± 0.13%). Not many PAHs are observed with the AMS for the plastic bags. The difference for the observed PAH contribution is mainly caused by the burning material, i.e., the precursor of PAHs, such as lignin, single-ring compounds, and aliphatic hydrocarbons. The burning of PE has lower yield of PAHs than lignin (Zhou et al., 2015), resulting in the lower PAH contribution in the burning of polyethylene plastic bags.

**3.2.2 Chemical composition of POAs measured with the EESI-TOF**

The EESI-TOF provides an important complement to the highly fragmented mass spectra generated by the AMS, where intact compounds measured by the EESI-TOF from $m/z$ 100 to 400 without assuming specific response factors toward each molecular formula are shown in Figure 3. The mass spectrum of plastic bags in not shown because the EESI-TOF is insensitive to hydrocarbons because of their low solubility in the electrospray and low affinity for $Na^+$. The bin of compounds containing O/C greater than 0.7 has the largest and similar contribution in wood burning (29.9% to 31.5%), and it is slightly smaller in straw (25%) and cow dung (20.1%). O/C smaller than 0.15 contributes 15.3% to 18.8% in spruce and pine which is similar to the fraction in cow dung (13.5%) but much higher compared to beech logs (4.7%) and straw (8.8%), mainly due to the greater contribution of compounds with carbon numbers in the range of 18 to 21. Cow dung has a slightly lower fraction of low-H/C and a slightly higher fraction of high-H/C comparing to other fuels studied.

As shown in the pie charts in Figure 3, the $C_xH_yO_z$-family is the main group measured by the EESI-TOF with contribution from 80% to 97%. The N-containing species have the highest contribution (19.6%) in the POA from cow dung open burning, which is much higher than other fuels in this study (2.5% to 8.9%). Of the N-containing species in cow dung POA 95% contain one nitrogen atom and are in a wide range of carbon number between 5 and 22. They are mainly in the O/C range of < 0.15 to 0.5 and the H/C from 1.2 to > 1.7 (Figure S4). The degree of unsaturation, calculated from the ratio of the double-bond equivalent to the number of carbons (DBE/C). The difference in all the studied POAs is not major (Figure S5).

On molecular level, $C_6H_{10}O_5$ ($m/z$ 162.0523) is most abundant in wood combustion (20.3% ± 6.1%), is less pronounced in straw (15.0% ± 7.9%) and even less so in cow dung emissions (9.9% ± 5.4%) (Figure 2c). It could be mainly assigned to levoglucosan (or similar dehydrated sugars) which is formed from the pyrolysis of cellulose and hemicellulose (Simoneit, 2002). The second most abundant species presented in the POAs is $C_8H_{12}O_6$ ($m/z$ 204.0628) in this study, contributing between 2% and 9%. It has been observed from the primary biomass burning emissions in the laboratory and ambient studies (Kumar et al., 2022; Kong et al., 2021). In addition, compounds with 18 and 20 carbon atoms are rich in many fuel types, particularly in spruce and pine burning emissions, and are notably minimal in beech logs.

The O/C (calculated as the ratio of total oxygen to total carbon) of the POAs from 5 types of burning measured by the EESI-TOF is 0.32 ± 0.07 to 0.41 ± 0.02, which is higher than that of the AMS (0.16 ± 0.07 to 0.37 ± 0.08). The difference likely occurs because the EESI-TOF is insensitive to species with low water solubility and/or low affinity for $Na^+$ (e.g., hydrocarbons including polyaromatic hydrocarbons). This will contribute to an underestimation of H/C. The total nitrogen-to-total carbon ratio (N/C) of cow dung measured by EESI-TOF is 0.019, which is slightly higher than that in the AMS measurements (0.015). This could be partially because of the difference in EESI sensitivity to the N-containing molecules. Another reason is that the total carbon from POA measured by the EESI-TOF is smaller, again consistent with the absence of non-water-soluble substances or molecules that do not bind to $Na^+$.

**3.3 Literature markers for solid-fuel combustion**

Levoglucosan and dehydrated sugars having the molecular formula $C_6H_{10}O_5$ are commonly used as tracers for biomass burning. A range of values for the fraction of $C_6H_{10}O_5$ is observed both for the same fuel type under different burning conditions and for different fuel types, as seen in Figure 2c. Thus, $C_6H_{10}O_5$ is a good untargeted

marker for biomass burning, but cannot be used to determine the specific source (or type of combustible) responsible for biomass burning emissions. Likewise, the $C_8H_{12}O_6$ is not a suitable marker for specific emission sources, as it is prevalent in all burns of biomass material. Additionally, $C_8H_{12}O_6$ has been considered as a tracer for terpene or syringol-derived SOA (Szmigielski et al., 2007; Yee et al., 2013), however our results suggest this molecular formula is not a good marker for SOAs due to the strong contribution from biomass burning-derived POA.

At a higher mass range, $C_{16}H_{32}O_2$ and $C_{18}H_{30,32,34}O_2$ are likely to be the common saturated and unsaturated fatty acids corresponding to palmitic, linolenic, linoleic, and oleic acid, which are important structural components of cells and have been found in the emission of cooking, biomass burning, and cow dung (Simoneit, 2002; Neves et al., 2009a; Neves et al., 2009b; Brown et al., 2021). The corresponding compounds for the $C_{20}H_{30}O_2$ and $C_{20}H_{28}O_2$ are most likely resin acids (e.g., abietic acid and pimaric acid) and dehydroabietic acid which have been specifically found in coniferous resin (Holmbom, 1977; Simoneit, 2002) and served as biomass burning tracers (Simoneit et al., 1993; Liang et al., 2021). The $C_{20}H_{30,32,24}O_{2,3}$ have been found as diterpenoids from the wood of Cunninghamia konishii (Li and Kuo, 2002). This plant species belongs to the class of Pinopsida, which also includes spruce and pine.

These typical markers stated above are well-known, but due to their presence in more than one fuel, the determination of different BB sources (or even biomass burning-derived POA) is challenging. For example, scaling levoglucosan to total BB OM requires a priori knowledge of the BB source and burning condition (Favez et al., 2010). Therefore, it is complicated to apply these markers in the source apportionment without comparison to statistically rule out other possibilities.

**3.4 Identification of potential markers for specific solid fuels**

To investigate the feasibility of distinguishing differences between the combustion fuel sources based on the measured species, the similarity of mass spectra acquired from each experiment by AMS and EESI-TOF is assessed with Spearman's rank correlation coefficient ($r$), as shown in Figure 4. The calculation of Spearman's coefficient is equivalent to calculating the Pearson correlation coefficient on the rank-ordered data, so it assesses monotonic relationships for ions from two mass spectra. In the correlation matrix with the fragment ions from AMS (Figure 4a), it is clear that the POAs from the same burning fuel strongly correlate. For instance, the average correlation coefficients of the AMS POA MS for all experiments using the same fuel range from 0.84 to 0.95. When comparing different fuels, a strong correlation is also found between spruce and pine logs stove burning and spruce and pine branches and needles open burning (0.95 ± 0.02). This is mainly because these two types of burning are closely related (i.e., derived from the same plants), and therefore have similar chemical composition. The correlation weakened when comparing POAs from different materials (e.g., spruce – beech 0.77 ± 0.03, spruce – straw 0.76 ± 0.03, spruce – cow dung 0.75 ± 0.03).

By contrast, the correlation coefficients based on the species from EESI-TOF are much lower among different burning fuels and even amongst the same fuel type (0.44 to 0.68). Noticeably, only a weak intra-fuel correlation is found for spruce/pine logs stove burning (0.44 ± 0.07), indicating that there are significant differences between experiments which are likely driven by burn-to-burn variability caused by differences in the combustion condition or variance of the fuel materials (e.g., with or without bark, amount of sap in the wood, etc.). Nevertheless, the variability between different fuels is clearly larger than the intra-fuel variability for the POAs. For example, the

correlation between the cow dung and all the other fuels (average $0.27 \pm 0.11$) is significantly lower than that of
among cow dung emissions ($0.49 \pm 0.16$). This suggests that the EESI-TOF may be capable of distinguishing
between different types of BB fuels.
To perform a more detailed analysis and identify markers between the emissions, the Mann-Whitney U test (see
Sect. 2.2) of the POAs from different fuels measured by AMS and EESI-TOF is conducted. Considering that both
spruce and pine logs stove burning as well as spruce and pine branches and needles are similar fuel types and have
a comparable POA composition in Figures 1 to 3, they were classified as the same fuel for this test. Results of the
Mann-Whitney U test are presented in Figure 5, where we show the average $-\log_{10}$ of the $p$-value as a function of
the $\log_2$ of the fold change (FC). Species having $p$-values less than 0.1 in the two-tailed test in all pairwise
comparisons are considered to be significantly more prevalent or scarcer in a single fuel compared to all other
fuels. These ions are represented as colored circles in Figure 5. If the species fail to meet the criterion one time or
more than one time, those species will be shown as gray circles even though their average $p$-value might be lower
than 0.1. A higher $-\log_{10}(p\text{-value})$ (i.e., a lower $p$-value) indicates a lower probability that the fractional medians
of two species are equal. At the same time, a higher FC (Equation 4) indicates a higher abundance of the species'
fractional contribution in the tested fuel compared to the average of all other fuels, deeming it more exclusive. In
the case of beech logs as well as spruce and pine logs burning, the colored $p$-value is lower (higher $-\log_{10}(p\text{-value})$)
in the dataset of AMS than that of EESI-TOF, suggesting the results from the AMS are more replicable. However,
from the perspective of FC, its absolute value is around 2 to 4 times higher in the dataset of the EESI-TOF than
that of the AMS. This shows that the potential markers selected from the EESI-TOF measurement are more unique,
in some cases found only in the spectrum of a given source. On this ground, the AMS and EESI-TOF are potent
complementary tools to provide separation and source apportionment of ambient OA, and to capture marker
compounds. The selected potential markers, $p$-values, and fold changes are listed in Table S2 and Table S3 for
EESI-TOF and AMS data, respectively.
Mass defect plots of the selected marker compounds are visualized in Figure 6. Many more potential markers are
identified from spruce and pine burning, as well as cow dung open burning, in comparison to beech and straw
burning. As shown in Figure 6A with the AMS dataset, potential markers from $C_xH_y$ and $C_xH_yO_z$-family have
significantly higher fraction in the POA of beech logs than those in other fuels. By contrast, the selected markers
for spruce and pine burning are more oxidized and mainly composed of $C_xH_yO_z$-family, which is consistent with
its bulk chemical composition and relatively higher O/C. The main fragments $CO^+$ and $CO_2^+$ have higher
contributions in spruce and pine burning (also can be seen in Figure 1), but their FCs are not very high, which
means they are not exclusive in spruce and pine and therefore are not applicable as sole tracers in the complex
ambient air. Fragments from cow dung open burning have considerably higher contribution in $C_xH_y$-family and
N-containing families, but lower in oxygen-containing species, which also agrees with bulk chemical composition
characteristics.
Similarly, many marker compounds are determined in the measurement of EESI-TOF for spruce and pine burning
as well as cow dung open burning. Compounds with $20 - 21$ carbon atoms as shown in Figure 6B for spruce and
pine burning could be resin and conifer needle-related, such as $C_{20}H_{32}O_3$ (likely isocupressic acid) (Mofikoya et
al., 2020; Wiyono et al., 2006). However, $C_{20}H_{30}O_2$ mentioned in previous section with notable abundance is not
stably emitted in each spruce and pine burn. Therefore, it is not determined as a marker for spruce and pine.
Homologues of $C_{11}H_{12}(CH_2)_{0-3}O_7$ are also determined, of which $C_{14}H_{18}O_7$ could be picein which is an important

phenolic compound in the needles of spruce (Løkke, 1990). On the other hand, some compounds which are barely present in the POA of spruce and pine burning, such as $C_{14}H_{28}(CH_2)_{0-3}O_2$ (likely saturated fatty acids), offers an alternative perspective of exclusion method in source separation. Noticeably, while coniferyl alcohol ($C_{10}H_{12}O_3$) is a major pyrolysis product from softwood (e.g., spruce) lignins (Saiz-Jimenez and De Leeuw, 1986) and has a decent fractional contribution in POA of the spruce and pine burning, but its contribution in spruce and pine burning is smaller than straw burning. Therefore, it is not recommended as a tracer when other biomass fuels are present. For the hardwood (i.e., beech logs in this study), sinapyl alcohol ($C_{11}H_{14}O_4$) is one of the prominent products from pyrolysis of lignins (Saiz-Jimenez and De Leeuw, 1986) and is conspicuous in our beech logs stove burning. Interestingly, nitrogen-containing compound $C_{13}H_{17}NO_6$ is noted as a tracer for straw open burning, and the nitrogen-containing fragments $C_3H_{8-9}N$ are also selected from the straw AMS analysis.

Cow dung is a clearly different fuel to other biomass fuels in this study, thus many markers are identified from cow dung open burning. These potential markers have mostly nitrogen in chemical composition and with generally higher FC. Many series of N-containing homologues are found, such as $C_{10}H_9NO_2$ and $C_{11}H_{11}NO_2$, which could be likely assigned to the derivative of indole, i.e., indole acetic acid and indolepropionic acid respectively. Another series of homologues is $C_9H_{11}NO_2$ and $C_{10}H_{13}NO_2$, which have been found especially in the emissions from cow dung cook fire in India compared to brushwood cook fire (Fleming et al., 2018). Homologues without nitrogen atoms in the chemical composition are also seen, for example, $C_{22}H_{42}(CH_2)_{0-2}O_2$, likely the homologues of erucic acid which is a natural fatty oil mainly present in the Brassicaceae family of plants. Nevertheless, it is not very surprising to see the biomass-related species as cows are herbivorous animals.

From the perspective of source-apportionment, ions that are primarily associated with a specific emission source and exhibit minimal contribution from other sources can be considered as potent in use. To show the ability of these markers for source separation, the contribution of two markers for the same source from Table S2 and Table S3 that possess small $p$-value with high FC are plotted among studied fuels. As shown in Figure S6, these markers measured by the AMS have relatively higher contribution in one specific fuel, which makes the fuel distinctive from others. Nonetheless, one would need to coordinate with more tracers to draw a conclusive diagnosis because the presence of these markers in other fuels. Given this scenario, the markers that have significantly low contribution (FC < 1) in a specific fuel could shed the lights on. In contrast, markers observed from the EESI-TOF is more robust for utilization as most of them are unique. As the markers listed in Table S2 and Table S3 are many, we could not thoroughly discuss here. However, due to the limited fuel types, referencing more markers can provide more confidence in source identification.

To the best of our knowledge, most of these markers are reported for the first time in POA emissions from the studied fuels with the EESI-MS, which is becoming more commonly used in the measurement of atmospheric aerosols. They could improve the refinement in source separation of fuels in biomass burning. Replicability and specificity are two important criteria for tracers. The $p$-value being less than 0.1 in the two-tailed test can ensure the stability of the shown results. The FC tells the degree of specificity of markers of one fuel in the presence of other fuels. If the $p$-value criterion is satisfied and the FC is large, the presence of this marker can directly lead us to the emission source. On the other hand, if the FC is less than 1, the detection of this compound can decisively exclude the related source after verifying this isn't due to a detection limit issue. However, if the FC is in-between, more caution is needed because these compounds don't have a distinctive fraction in that fuel compared to other fuels, but could have a relatively fixed ratio compared to other markers.

**4 Conclusions**

In this study, we conducted 36 burning experiments to simulate typical solid fuel combustion emission, including residential burning (beech or spruce and pine logs stove burning), wildfire (spruce and pine braches and needles open burning), and agricultural residue in field burning (straw open burning), cow dung open burning, and plastic bags open burning. The emission factors of CO, $CO_2$, THC, PM, OM, and BC were determined. The chemical composition of particles emitted from the combustion processes was comprehensively characterized using the AMS and the EESI-TOF, and the chemical composition of the particles measured by the two instruments were compared. These are the first direct measurements of these source profiles with the EESI-TOF. The utility of traditional markers are discussed, and new potential markers were identified using the Mann-Whitney U test.

The EFs of CO and THC are generally higher during the low combustion efficiency, and the opposite for the EF of $CO_2$. The highest EF of PM ($16.6\pm10.8$ g kg$^{-1}$) is from cow dung open burning which is mostly OM ($16.2\pm10.8$ g kg$^{-1}$), but for residential and plastic bags burning, the eBC accounts for ~30% of the total PM. The organics measured by the AMS show that the wood (beech, spruce, and pine) burning emission has a relatively higher abundance of $C_xH_yO_z$ fragments, while straw and cow dung burning emissions are dominated by $C_xH_y$ fragments in their POAs. On the molecular level, $C_6H_{10}O_5$ has the highest proportion (~7% to ~30%) in the POAs measured by the EESI-TOF (except for the plastic bags burning), followed by $C_8H_{12}O_6$ with fractions of ~2% to ~9%. The chemical composition measured with AMS covers a wide range of non-refractory organic and inorganic components. However, the extensive fragmentation concentrates the measured mass-to-charge ratio below ~120 and limits its chemical resolution. The chemical groups used to deduce the composition of particles could originate from different compounds, which impedes us from seeing the full picture. The formula-based mass spectrum from the EESI-TOF overcomes this deficiency and thus reveals the detailed characteristics.

However, many compounds are present ubiquitously in all of the fuels used here, making it challenging to identify atmospheric sources solely by visual comparisons of the full mass spectra. By using the Mann-Whitney U test to identify potential markers among the studied fuels, we find that the markers identified by the AMS have greater replicability and by EESI-TOF are more distinctive, thus providing an important reference for the source apportionment. Overall, this work highlights the complex characteristics of POAs emitted from the burning of solid fuels and proposes the markers for separating different sources using the AMS and EESI-TOF. This work shows mass spectral profiles of burning emissions on bulk and molecular level, which improves our understanding of POA from different fuels. The markers provided in this study are crucial for distinguishing the sources of aerosols in the atmosphere and enhancing the interpretation of source apportionment. In the future, the volatility and chemical reactivity of the proposed markers should be tested to determine their atmospheric stability and their ability to be a robust marker. More burning fuels such as coal and grass could be conducted to enrich the spectral database.

Future studies will probe the usefulness of these markers, if they are long lived enough in the atmosphere to provide useful separation between complex emission sources shown here. This will either focus on online measurements in polluted regions or from offline filter analysis from similar regions. Clearly, the dominant biomass burning markers (levoglucosan and others) are not robust to be used to separate different biomass sources, though they are robust for identification of general biomass burning aerosol. Nitrogen containing compounds emitted from cow dung emissions can provide a very unique set of markers for separating this source from other

biomass sources. Additionally, resin acids from observed in the emissions from spruce and pine emissions provide
unique species associated with these emissions (and observed previously).
At the present moment, to provide insight into the usefulness of these markers within the context of ambient
measurements, or against source apportionment methods, we would require a robust dataset of comparable data
to test these markers and average emission profiles against.
**Data availability**
The datasets are available upon request to the corresponding authors.
**Author contributions**
JZ, TTW, KL, DMB, EG, KYC, and SB conducted the burning experiments. JZ analyzed the data and wrote the
manuscript. DSW, MS, TQC, LQ, and DB participated in the campaign. DMB, KL, IEH, HL, JGS, and ASHP
participated in the interpretation of data.
**Competing interests**
The authors declare that they have no conflict of interest.
**Disclaimer**
Publisher's note: Copernicus Publications remains neutral with regard to jurisdictional claims in published maps
and institutional affiliations.
**Acknowledgements**
This work was supported by the Swiss National Science Foundation (SNSF) SNF grant MOLORG
(200020_188624), the European Union's Horizon 2020 research and innovation programme through the ATMO-
ACCESS Integrating Activity under grant agreement no. 101008004, the European Union's Horizon 2020
research and innovation programme under the Marie Skłodowska-Curie grant agreement No. 884104 (PSI-
FELLOW-III-3i), SNSF Joint Research Project (No. 189883) and grant No. 206021_198140.

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

    **Tables and figures**

    **Table 1 Average emission factors of CO, CO$_2$, THC, PM, OM, and BC as well as MCE for 6 types of burning.**

| Burning type | Carbon content | Particle density (g cm$^{-3}$) | MCE | Emission factors (g kg$^{-1}$ fuel) | | | | | |
|---|---|---|---|---|---|---|---|---|---|
| | | | | CO | CO$_2$ | THC | PM | OM | eBC[*] |
| beech logs stove (n=5) | 0.46 | 1.70 | 0.91±0.03 | 85.8±25.9 | 1466.9±65.8 | 19.3±5.5 | 7.6±2.2 | 6.2±2.8 | 2.43±0.9 |
| spruce and pine logs stove (n=8) | 0.46 | 1.70 | 0.91±0.02 | 83.8±26.7 | 1640.7±58 | 16.1±4.8 | 4.9±2.2 | 2.0±1.3 | n.a |
| spruce and pine branches and needles open (n=4) | 0.46 | 1.70 | 0.93±0.02 | 63.5±6.8 | 1668.9±26.7 | 14.1±3.4 | 9.4±2.7 | 3.8±1.1 | n.a |
| straw open (n=6) | 0.45 | 1.50 | 0.95±0.04 | 44.4±34.1 | 1511.7±103.2 | 19.1±17 | 2.8±1.2 | 2.4±1.3 | 0.7±0.2 |
| cow dung open (n=6) | 0.45 | 1.54 | 0.87±0.03 | 92.3±27.4 | 1366.2±88.4 | 30.3±8.5 | 16.6±10.8 | 16.2±10.8 | 0.8±0.3 |
| plastic bags open[**] (n=4) | 0.84 | 0.45 | 0.98±0.02 | 29.3±39.2 | 2956.6±138.9 | 18.2±28.6 | 2.7±1.2 | 1.1±0.3 | 1.0±0.3 |

    [*] The number of burns is indicated by n. BC data is not available for some burns.

    [**] The emission factors of PM and OM are corrected as explained in the Supplement.

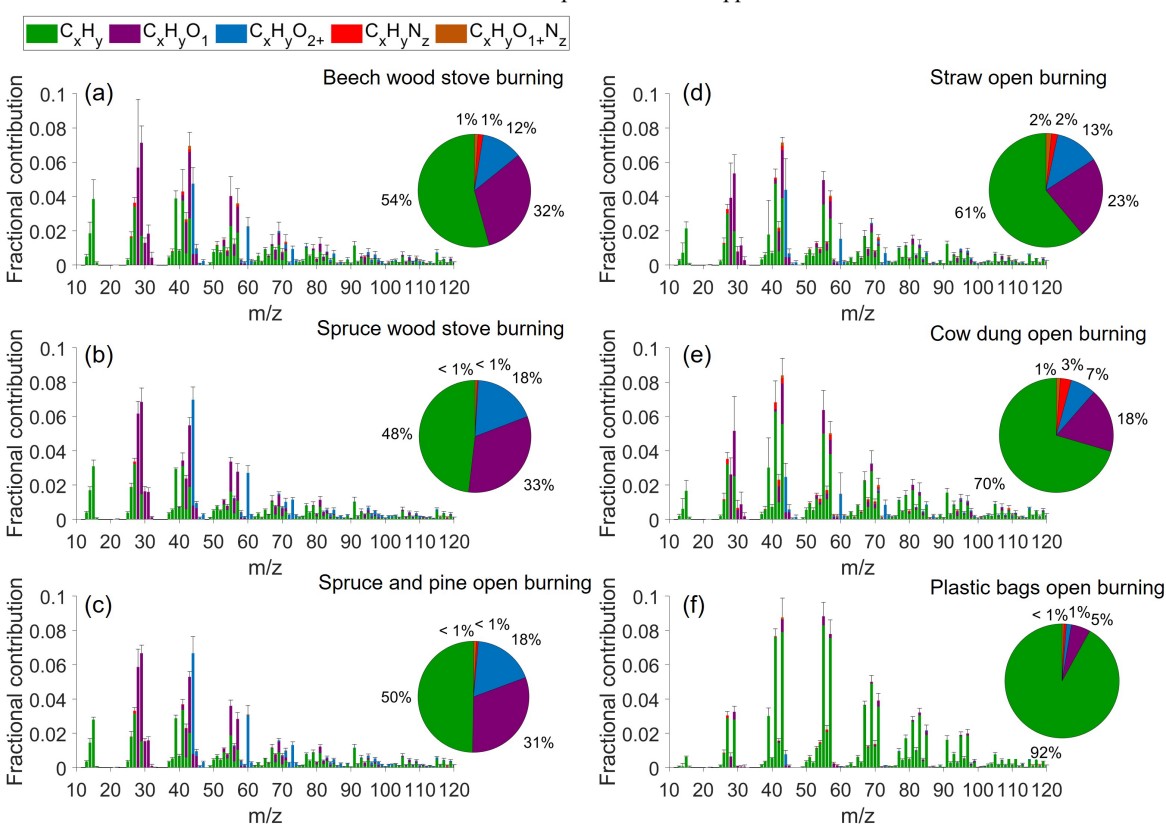

    **Figure 1 Average AMS POA mass spectral profiles and elemental compositions of (a) beech logs stove burning (n=6; n**
    **is the number of experiments), (b) spruce and pine logs stove burning (n=9), (c) spruce and pine branches and needles**
    **open burning (n=4), (d) straw open burning (n=6), (e) cow dung open burning (n=5), and (f) plastic bags burning (n=4).**
    **The error bar denotes half standard deviation in grey. The pie chart showing the contribution of elemental families is**
    **at the right of the mass spectrum.**

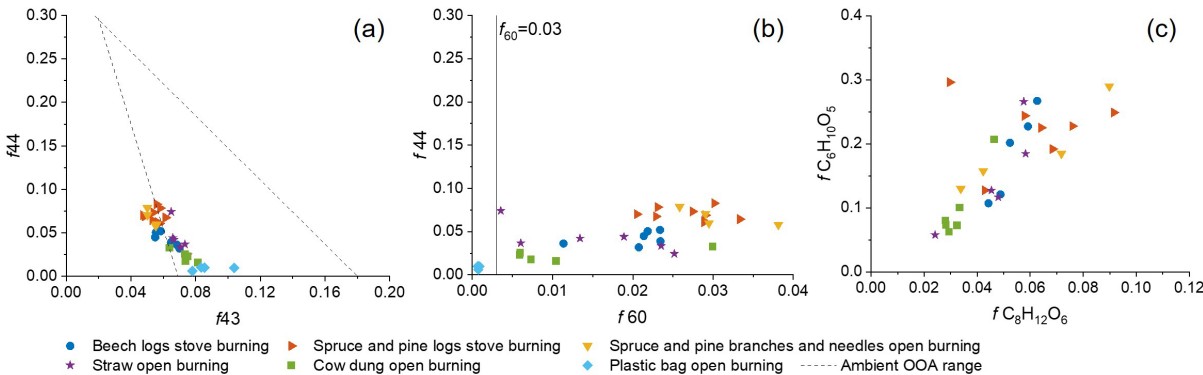

774

**Figure 2. Scatter plots of (a) $f_{44}$ vs. $f_{43}$ from AMS, (b) $f_{44}$ vs. $f_{60}$ from AMS, and (c) $f$ C₆H₁₀O₅ vs. $f$ C₈H₁₂O₆ from EESI-TOF. The dashed line denotes the estimated OOA range and the solid line denotes $f_{60}$ background level in the ambient from Ng et al. (2010) and Cubison et al. (2011).**

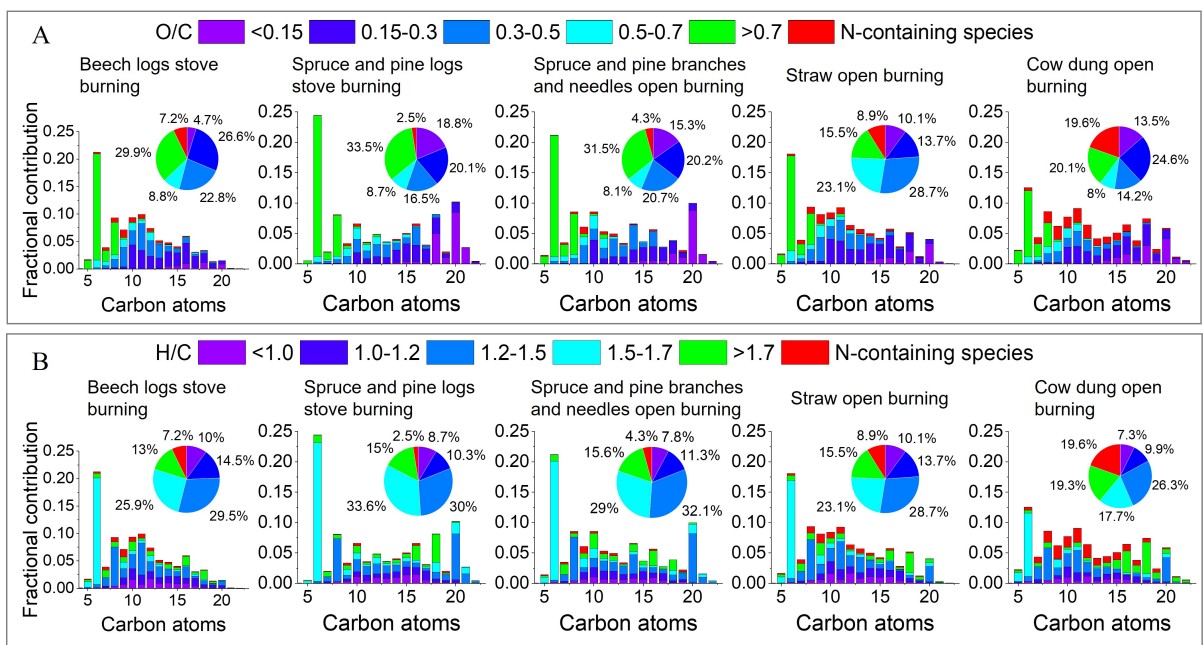

778

**Figure 3 The average carbon and oxygen distribution colored by the O/C and H/C for non-N-containing species in panel A and B respectively with EESI-TOF. The N-containing species are colored in red. The pie charts are the corresponding contribution of a range of O/C or H/C ratios.**

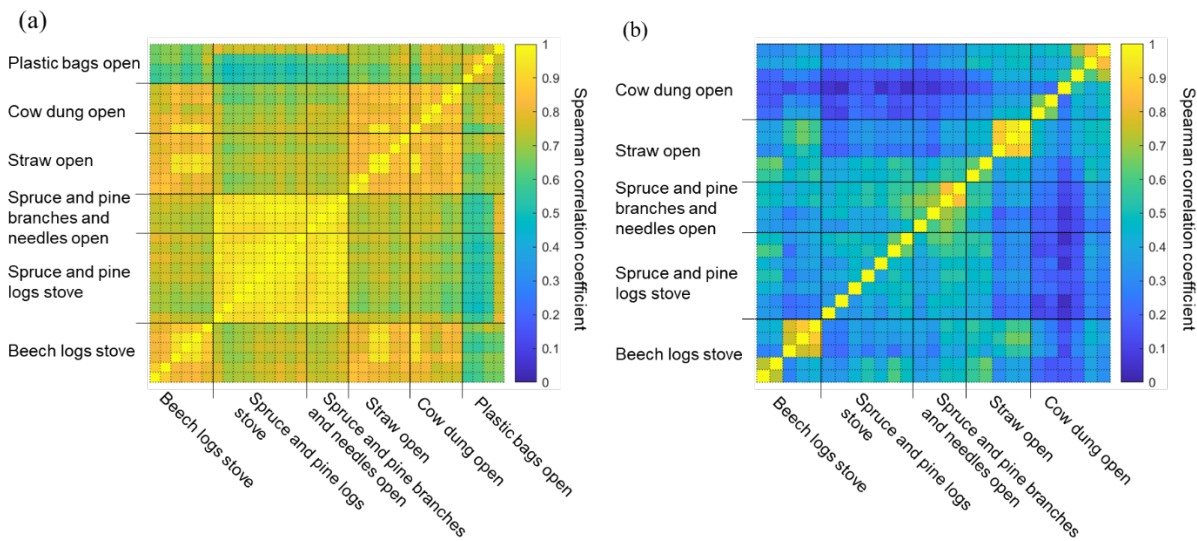

782

**Figure 4 The correlation matrix of POAs measured with (a) the HR data from AMS and (b) EESI-TOF using Spearman correlation function. Note that some experiments did not have either AMS or EESI-TOF data.**

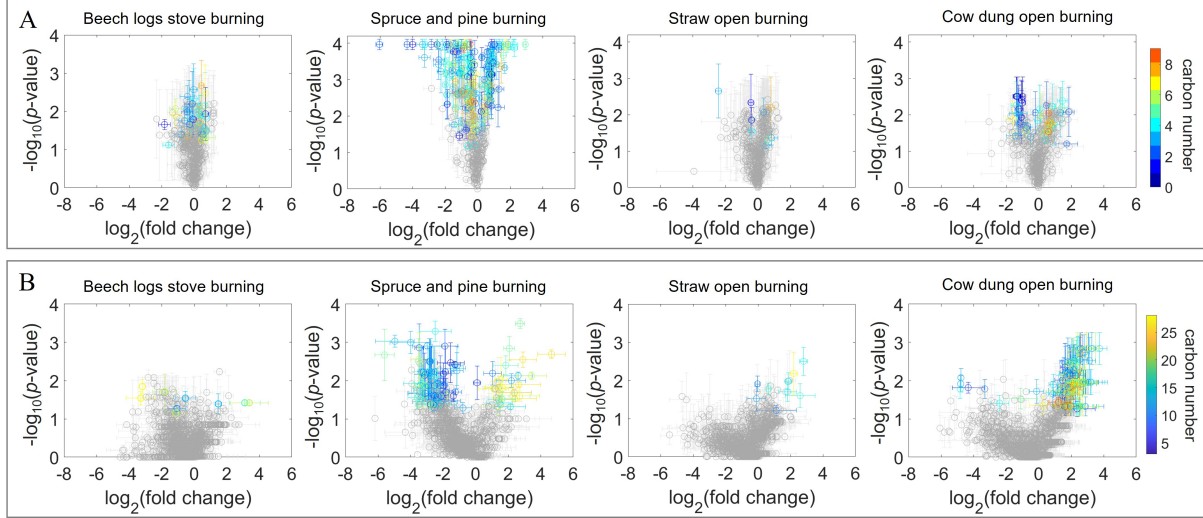

**Figure 5 The statistic *p*-value vs. fold change with the dataset from AMS in panel A and EESI-TOF in panel B. The color bars are the number of carbon atoms. The horizontal error bars are the 1 standard deviation given by the *p*-value variations in the pairwise tests, and the vertical error bars are the 1 standard deviation of the $log_2$(fold change).**

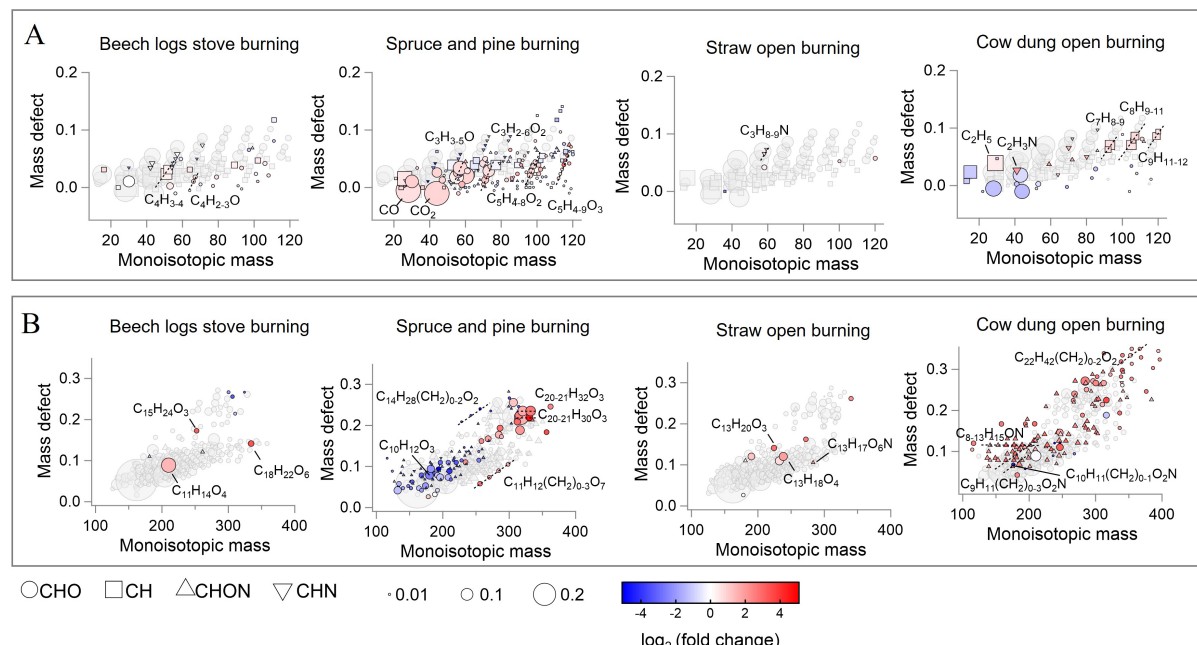

**Figure 6 The mass defect plot with the dataset from AMS in panel A and EESI-TOF in panel B. The markers denote the fragments or molecules having the same formula. They are sized by the square root of fractional contribution and colored by the $log_2$(fold change).**