# Peer review of "Bulk and molecular-level composition of primary organic aerosol from wood, straw, cow dung, and plastic burning"

_EGUsphere, 2023_

## Author Comment (AC1)

Reviewer #1

Zhang et al. used advanced mass spectrometry tools to measure the primary organic aerosol emitted from burning a variety of biomass fuels and plastic. They reported emission factors of gases and organic aerosol from burning these fuels, and used EESI-ToF measurement to identify unique tracers for different fuels with solid statistical analysis. The very new instrument, EESI-ToF, is used to characterize biomass burning particulate matter for the first time. Given the expanding use of EESI-ToF by the community, I think this work fits the scope of ACP. The manuscript is very clearly written. I recommend this work be published after some minor changes. I have the following comments for the authors to consider.

We would like to thank the reviewer for the comments and suggestions to improve the current work. We will have the reviewer comments in black, address the comments in blue, and modified sentences in blue and italics.

Line 21 in the abstract: From the main text, I found out that the organic gases were measured by a total hydrocarbon analyzer. However, when reading the abstract, I thought that they were measured by AMS or EESI. I would suggest the authors talk about how the hydrocarbon gases were measured before reporting these numbers.

Response: The instruments used for the measurements are added as follows (line 20 to 22).

"*The emission factors of organic matter estimated by AMS and hydrocarbon gases estimated by the total hydrocarbon analyzer are $16.2 \pm 10.8$ g kg$^{-1}$ and $30.3 \pm 8.5$ g kg$^{-1}$ for cow dung burning*"

Line 25 in the abstract: I don't fully understand what this sentence means. The authors may want to rewrite it.

Response: It has been deleted because PAHs are not the main scope of this paper.

Line 112: Does "THC" here and in Equation 2 include methane?

Response: Yes, THC includes methane as defined throughout this paper. Line 112 in the original manuscript described parallel THC and $CH_4$ because Horiba APHA-370 measures THC and $CH_4$ using different channels and displays both concentrations. The calculation of emission factors using the carbon mass balance method needs the overall carbon mass concentration as the denominator, therefore the THC in Equation 2 also includes methane.

Considering that methane data is not used alone in this paper, we remove the methane from Line 112 to avoid misunderstanding as follows on line 120:

"*Gas analyzers were used for monitoring the concentration of CO (Horiba APMA-370), $CO_2$ (LI-COR LI-7000), and total hydrocarbon (THC, including methane) (Horiba APHA-370).*"

Line 115&117: The mass resolution of the AMS instruments can be documented here. Same for the EESI-ToF below.

Response: The mass resolution of EESI-TOF has been shown in Line 131 in our original manuscript as follows:

"*The EESI-TOF mass analyzer achieved a mass resolution of ~10000 at m/z 173 and 11000 at m/z 323.*"

The mass resolutions of the AMS instruments are added as follows from line 123 to line 127:

"*A long time-of-flight aerosol mass spectrometer (LTOF-AMS, Aerodyne Research, Inc.) with a mass resolution of ~5000 over the range of m/z 100 to m/z 450 deployed for online, non-refractory particle characterization and a subset of experiments were performed with high-resolution time-of-flight AMS (HTOF-AMS, Aerodyne Research, Inc.) with a mass resolution of ~2000 over the range of m/z 100 to m/z 450.*"

Line 138: In Equation 1, as I understand, MassX is the mass flux to the detector, and the authors are not attempting to quantify the compounds. Did the authors assume that all the compounds have the same response factor when making Figure 3?

Response: Yes, we assume that all the measured compounds have the same response due to the lack of commercial standards. The response factor was discussed for secondary organic aerosols (Wang et al., 2021), but so far it is not available for primary organic

aerosols. Intercomparisons with the AMS have shown that the total EESI-TOF signal responds quantitatively to the bulk organic aerosol concentration (Lopez-Hilfiker et al., 2019; Wang et al., 2021).

It has been clarified as follows from line 258 to line 260.

*"The EESI-TOF provides an important complement to the highly fragmented mass spectra generated by the AMS, where intact compounds measured by the EESI-TOF from m/z 100 to 400 without assuming specific response factors toward each molecular formula are shown in Figure 3."*

Line 154: Would be great to document the (average) densities of aerosol in Table 1 or Table S1.

Response: The densities of particles in the burning emissions are added in Table 1.

*Table 1 Average emission factors of CO, $CO_2$, THC, PM, OM, and BC as well as MCE for 6 types of burning.*

| Burning type | Carbon content | Particle density (g cm⁻³) | MCE | Emission factors (g kg⁻¹ fuel) | | | | | |
|---|---|---|---|---|---|---|---|---|---|
| | | | | CO | $CO_2$ | THC | PM | OM | eBC[*] |
| beech logs stove (n=5) | 0.46 | 1.70 | 0.91±0.03 | 85.8±25.9 | 1466.9±65.8 | 19.3±5.5 | 7.6±2.2 | 6.2±2.8 | 2.43±0.9 |
| spruce and pine logs stove (n=8) | 0.46 | 1.70 | 0.91±0.02 | 83.8±26.7 | 1640.7±58 | 16.1±4.8 | 4.9±2.2 | 2.0±1.3 | n.a |
| spruce and pine branches and needles open (n=4) | 0.46 | 1.70 | 0.93±0.02 | 63.5±6.8 | 1668.9±26.7 | 14.1±3.4 | 9.4±2.7 | 3.8±1.1 | n.a |
| straw open (n=6) | 0.45 | 1.50 | 0.95±0.04 | 44.4±34.1 | 1511.7±103.2 | 19.1±17 | 2.8±1.2 | 2.4±1.3 | 0.7±0.2 |
| cow dung open (n=6) | 0.45 | 1.54 | 0.87±0.03 | 92.3±27.4 | 1366.2±88.4 | 30.3±8.5 | 16.6±10.8 | 16.2±10.8 | 0.8±0.3 |
| plastic bags open[**] (n=4) | 0.84 | 0.45 | 0.98±0.02 | 29.3±39.2 | 2956.6±138.9 | 18.2±28.6 | 32.7±1.2 | 1.1±0.3 | 1.0±0.3 |

[*] *The number of burns is indicated by n. BC data is not available for some burns.*
[**] *The emission factors of PM and OM are corrected as explained in Supplement.*

*"The densities could be underestimated because of the non-spherical shape of particles, especially particles from plastic bags burning mainly due to the high contribution of BC."* (Line 164 to line 166)

Line 204: The CHON ions are not included in Figure 1. Is it because they are negligible? Were they fitted in Pika?

Response: Yes, the fragments containing $CHO_1N$ and $CHO_{2+}N$ are fitted in Pika, but their individual contribution is less than 1%. We added the $C_xH_yON_z$ family in Figure 1 for the concern.

[Figure]

*Figure 1 Average AMS POA mass spectral profiles and elemental compositions of (a) beech logs stove burning (n=6; n is the number of experiments), (b) spruce and pine logs stove burning (n=9), (c) spruce and pine branches and needles open burning (n=4), (d) straw open burning (n=6), (e) cow dung open burning (n=5), and (f) plastic bags burning (n=4). The error bar denotes half standard deviation in grey. The pie chart showing the contribution of elemental families is at the right of the mass spectrum.*

Line 206: If the ions in the CxHyOz family in the plastic burning spectrum are coming from burning other fuels, then are PM, OM, and BC emission factors reported for burning plastic bags still reliable? Also, did the authors see CxHyOz in all four plastic bag burning experiments?

Response: We observed a high contribution of $C_xH_yO_z$ in 3 out of 4 plastic burning experiments. We accordingly added a corresponding section in the SI where we estimate the contribution of the contamination of the mass spectra of plastic bags. Please find the details we added in the SI on line 11 to line 23 as below. The emission factor has been corrected for the relevant experiments in Table S1 and Table 1.

"***Plastic bags burning emission correction.*** *In three out of four plastic bags burning experiments, the mass spectrum at the middle to end burning stages had considerable $C_xH_yO_z$ family contribution (~23%). As the combustion progressed, the chimney was heated, and the volatile substances remaining on the chimney evaporated and were then partitioned to the particles for detection. However, at the early stage, before the chimney got hot, the mass spectra consisted mainly of hydrocarbons (see Figure S2a). Therefore, we take only the early burning stage of these three burning experiments into account for the average mass spectrum in Figure 1(f). The absolute concentration of the three AMS mass spectra derived from the early-stage burning is scaled to the uncontaminated burning experiment ions based on m/z 81 and m/z 83, which are stable and characteristic for hydrocarbons. The difference on average is 0.4% ± 1.0% which is very minor as shown in Figure S2b. The mass spectra of three contaminated burning over the whole burning stages indicate that the measured organics was 14.6% ± 8.7% overestimated. Correspondingly, the emission factors for PM and OM are corrected for each plastic bags burning.*

[Figure]

*Figure S2 (a) The time series of some ions measured by the AMS during the plastic bags burning for the contaminated case; (b) the mass spectrum comparison of uncontaminated plastic bags burning experiment at the top v.s. the average of 3 early-stage burning at the bottom.*"

Line 219: What is the "$f_{60}$ filter"? Is it 0.003?

Response: The $f_{60}$ filter is 0.003. The original sentence is modified on line 232 as below.

"*..., suggesting the $f_{60}$ filter ($f_{60}$ = 0.003) in the ambient is unlikely to miss biomass combustion.*"

Line 229: It might be good to label the PAH ions in the mass spectra (Figure S2).

Response: The relevant ions have been marked in Figure S2.

[Figure]

***Figure S2. Average AMS POA mass spectral profiles in the range from m/z 120 to 450 of (a) beech logs stove burning (n=6; n is the number of experiments), (b) spruce and pine logs stove burning (n=9), (c) spruce and pine branches and needles open burning (n=4), (d) straw open burning (n=6), (e) cow dung open burning (n=5), and (f) plastic bags open burning (n=4). The m/z for some ions are marked in the figure. The error bar denotes half standard deviation in grey.***

Line 230 and 231: The authors may want to double-check what the ion with m/z 239 is. It should not be the parent ion of a hydrocarbon. Also, is the molecular formula of methylbenzofluoranthene C19H12? I think it should be C20H14. I am also curious

whether the authors found a hint of retene in the AMS mass spectra because retene is usually a very abundant PAH emitted from burning conifers. It could have fragmented into smaller ions given its branched structure.

Response: The *m/z* 239 could be methylbenzo[ghi]fluoranthene or fragmentation of dehydroabietic acid according to the literature. The parent ion of retene at *m/z* 243 does not have a significant signal due to the fragmentation, but the main fragment of it at *m/z* 219 was found in the spruce and pine burning experiment.

Please find the modified sentences below from line 243 to line 250..

"*..., parent ions at m/z 226, 252, 276, 300, and 326 correspond respectively to $C_{18}H_{10}$ (benzo[ghi]fluoranthene and cyclopenta[cd]pyrene) $C_{20}H_{12}$ (benzofluoranthene and benzopyrene), $C_{22}H_{12}$ (indenopyrene and benzoperylene), $C_{24}H_{12}$ (coronene), and $C_{26}H_{14}$ (dibenzoperylene). The fragment of m/z 239 could be methylbenzo[ghi]fluoranthene ($C_{19}H_{12}$) (Dzepina et al., 2007; Ji et al., 2010) or a fragment of dehydroabietic acid which has been found in fresh pine resin (Colombini et al., 2005). The m/z 219 and m/z 285 also could arise from the fragmentation of retene and dehydroabietic acid, respectively, which also can be derived from conifer resin (Dzepina et al., 2007; Jen et al., 2019; Zetra et al., 2016).*"

Line 262: Compounds with 18-20 carbon atoms could be resin acids (or their decomposition products in biomass burning), which are abundant in conifers. Also, in Line 286, I am curious that did the authors see emission of dehydroabietic acid (C20H28O2) from burning coniferous fuels?

Response: Yes, we observed $C_{20}H_{28}O_2$. We have adapted this in the relevant description as below from line 305 to line 308.

"*The corresponding compounds for the $C_{20}H_{30}O_2$ and $C_{20}H_{28}O_2$ are most likely resin acids (e.g., abietic acid and pimaric acid) and dehydroabietic acid, respectively, which have been found in coniferous resin (Holmbom, 1977; Simoneit, 2002) and suggested as biomass burning tracers (Simoneit et al., 1993; Liang et al., 2021).*"

Line 298: How is the correlation coefficient calculated? The authors may want to provide more details.

Response: The details of correlation coefficient calculation have been added in the manuscript as follows on line 319 to 321:

"*The calculation of Spearman's coefficient is equivalent to calculating the Pearson correlation coefficient on the rank-ordered data, so it assesses monotonic relationships for ions from two mass spectra*."

Line 409: I am convinced by this analysis that the p-value and FC methods can select tracer compounds from different biomass burning fuels very efficiently. However, in source apportionment studies, there are usually non-biomass burning PM sources. I would suggest that the authors compare the spectra of the biomass burning POA with OA from other common sources in their future study or verify these tracers in future field campaigns to make sure they are exclusively from biomass burning.

Response: Thank you for the forward-looking approach that this method can offer. Indeed we plan to compare these specific markers noted in the future to evaluate the utility of tracers in combination with FC and *p*-value within the range of studied fuels. Certainly, if we can develop mass spectrum data in a wide range of sources in the future, we can make better use of this method and reduce the uncertainty in its application. Additionally, we plan in the future to assess these markers against secondary organic aerosol formation from the same emissions to assess their stability and continued uniqueness.

**Minor Comments**

Line 251-253: This seems to be an unfinished sentence.

Response: it is modified as follows on line 270 to 272.

"Of the N-containing species in cow dung POA 95% contain one nitrogen atom and are in a wide range of carbon number between 5 and 22. They are mainly in the O/C range of < 0.15 to 0.5 and the H/C from 1.2 to > 1.7 (Figure S4)."

Line 257: A redundant "%" should be removed.

Response: It is removed from the manuscript.

Line 696: "markers denote"

Response: It is changed as suggested.

---

## Author Comment (AC2)

Reviewer #2

**General comments:**

This study estimated the emission factors and characterized the POA with AMS and EESI-TOF from a variety of solid fuels. This topic is interesting and the experiment is well-designed.

However, some typos and misunderstandings were found. More necessary discussions and details are suggested to be provided. The authors should provide more convincing results. I recommend a major revision of this manuscript.

We thank the reviewer for the helpful comments. Below we provide a detailed point-by-point response to the issues raised by the reviewer. We will have the reviewer comments in black, address the comments in blue, and modified sentences in blue and italics.

**Major concerns**

The abstract part does not stress the importance of this manuscript, only emphasizing that OA is important, biomass burning is the common source of OA, and EESI-TOF is powerful. What is the urgency of measuring OA from the molecular level? What is the breakthrough and highlight of this work?

Response: As we have shown in the abstract, the urgency for molecular-level characterization of particulate matter stems from the inadequacy of chemical fragments to identify particulate matter from similar sources (Line 13-15). Such chemical separation is necessary to provide insight into the sources of air pollution in developing, and polluted, regions. The breakthrough and highlights of this paper are building up the molecular-level mass spectra from burning emissions and providing robust marker for source apportionment to assess the importance of different burning processes. Future measurements in such regions will provide insight into the ability of these markers to assess the importance of different biomass burning sources.

Please add some instrument comparisons in the introduction part. Why is EESI-TOF important? We admit that less decomposition or fragmentation is observed in EESI-TOF measurement, however, the homolog speciation could not be achieved.

Response: More instrument comparison has been added as below on line 76 to line 78.

"*Liquid chromatography-mass spectrometer can avoid thermal desorption and separate mixtures including isomers based their chemical affinity with the mobile and stationary phases (Zhang et al., 2021). However, it requires pre-treatment of samples which could introduce artefacts and lowers the time resolution*."

Is burning plastic bags really important? Please add more details or figures to illustrate this.

Response: The importance of burning plastic bags has been added in the manuscript as follows from line 46 to line 50.

"*Plastic burning has been estimated to contribute 13.4% of fine particulate matter ($PM_{2.5}$) yearly in India (Gadi et al., 2019), 6.8% in wintertime in China (Haque et al., 2019), and 2% to 7% in wintertime in the US (Islam et al., 2022). The toxic pollutants released from plastic burning, including olefins, paraffin, and polycyclic aromatic hydrocarbons, can cause respiratory irritation, and carcinogenic and mutagenic effects (Pathak et al., 2023).*"

The standard deviation of CO, and THC of straw burning and plastic bag burning is extremely high. Please specify. Note that other pollutants are not varied significantly.

Response: There is inherent variability in burn-to-burn. The high standard deviation of CO, $CO_2$, and THC for straw and plastic bag burning is driven by the combustion efficiency, i.e. MCE. The more flaming (MCE close to 1) the fire is the more emission of $CO_2$ and less emission of CO and THC. As shown in Table S1, the most flaming experiment of "SO4" with MCE of 1 has the largest emission factor (EF) of CO (97.4 g kg$^{-1}$) and smallest EF of THC (1366.1 g kg$^{-1}$) in all straw burning compared to the experiment of "SO6" with MCE of 0.89 (CO EF 0.2 g kg$^{-1}$, THC EF 1636.4 g kg$^{-1}$). For the same reason, the standard deviation is also high for plastic bags.

It has been specified from line 196 to line 198 as below.

*"Taking straw burning as an example, as shown in Table S1, the EFs of THC vary from 0.7 to 39.3 g kg$^{-1}$, with the MCE varying from nearly 1.00 to 0.89 correspondingly, resulting in the high standard deviation of the EFs."*

Line 229- 239: please add more detail or discussion about WHY PAHs are so different among samples. Does the material or burning styles shape the emission pattern?

Response: To begin with, please note that we corrected the mass spectra for plastic bags as mentioned in Reviewer #1's comments. The updated spectra and PAH peaks are marked in the new Figure S3, and show less difference than the old figure between fuel types.

The burning fuel and combustion efficiency are both important for the PAHs emission, as PAHs are the pyrolysis products of the burning material. However, the burning style used in this study does not cause a clear difference for PAHs, because the PAHs are very similar between spruce and pine logs stove burning (Figure S2b) and spruce and pine branches and needles open burning (Figure S2c). We note the lack of PAHs in cow dung emissions, which were dominated by smoldering conditions given their low MCE (0.87), nevertheless this difference may still be due to the fuel type.

More discussion has been added to our manuscript from line 252 to line 256.

*"Not many PAHs are observed with the AMS for the plastic bags. The difference in the observed PAH contribution is mainly caused by the burning material, i.e., the precursor of PAH, such as lignin, single-ring compounds, and aliphatic hydrocarbons. The burning of PE has a lower yield of PAHs than lignin (Zhou et al., 2015), resulting in the lower PAH contribution for polyethylene plastic bags."*

Line 264-267: add more uncertainty analysis to the quantitative or qualitative measurement.

Response: The standard deviation has been added as below from line 283 to line 284.

*"The O/C (calculated as the ratio of total oxygen to total carbon) of the POAs from 5 types of burning measured by the EESI-TOF is 0.32 ± 0.07 to 0.41 ± 0.02, which is higher than that of the AMS (0.16 ± 0.07 to 0.37± 0.08)."*

Line 273, how could the authors be sure that $C_6H_{10}O_5$ (m/z 162.0523) is levoglucosan (or similar dehydrated sugars)? Is there any GC-MS measurement?

Response: There are no GC-MS measurements associated with our measurements here, but we rely upon the breadth of previous work associated with quantification of the importance of dehydrated sugars (e.g. levoglucosan) present in many different biomass burning emission sources (Fabbri et al., 2008; Engling et al., 2006; Pashynska et al., 2002).

Part 3.3, Levoglucosan is not a good marker. However, $C_{16}H_{32}O_2$ is also not a good marker. Note that cooking emissions could also result in emissions of these so-called "markers" in this work. Please add more details.

Response: Section 3.3 aims to discuss and compare the often-mentioned markers in literature with our measurements instead of defining good markers. To better clarify this, the heading of the section has been changed to "Literature markers for solid-fuel combustion".

These markers are not defined as markers for the studied fuels. The "good" or "not good" marker is difficult to define without conditions. A detailed discussion of markers for specific fuels can be seen in Section 3.4. As the manuscript says, "the $C_6H_{10}O_5$ is a good untargeted marker for biomass burning" and it has been widely used in source apportionment. The cooking source of $C_{16}H_{32}O_2$ in our manuscript has been mentioned in Line 284 in the original manuscript.

Line 370: how could the authors be convinced that most of these markers are reported for the first time in POA emissions from the studied fuels? The investigation of pyrolysis of lignins has lots of results related to sugars, alcohols, and benzaldehydes.

Response: Due to the ubiquitous presence of lignins in the biomass materials, the pyrolysis products of it are less likely to be selected as useful markers for specific fuels. While some of the same markers may have been shown for previous biomass burning results as the chemical composition, our study is the best of our knowledge the first to determine the markers' ability to distinguish between specific fuels, as well as the first demonstration of the EESI-MS in conjunction with organic aerosol produced from biomass burning emissions.

The sentence has been modified as below from line 405 to line 406.

*"To the best of our knowledge, most of these markers are reported for the first time in POA emissions from the studied fuels with the EESI-MS, which is becoming more commonly used in the measurement of atmospheric aerosols."*

Please add more atmospheric implications in the conclusion part. The current paragraphs are only the common conclusions stated by the authors before.

Response: The implications have been added to the conclusion section from line 441 to line 458.

*"This work shows mass spectral profiles of burning emissions on bulk and molecular level, which improves our understanding of POA from different fuels. The markers provided in this study are crucial for distinguishing the sources of aerosols in the atmosphere and enhancing the interpretation of source apportionment. In the future, the volatility and chemical reactivity of the proposed markers should be tested to determine their atmospheric stability and their ability to be a robust marker. More burning fuels such as coal and grass could be conducted to enrich the spectral database.*

*Future studies will probe the usefulness of these markers, if they are long lived enough in the atmosphere to provide useful separation between complex emission sources shown here. This will either focus on online measurements in polluted regions or from offline filter analysis from similar regions. Clearly, the dominant biomass burning markers (levoglucosan and others) are not robust to be used to separate different biomass sources, though they are robust for identification of general biomass burning aerosol. Nitrogen containing compounds emitted from cow dung emissions can provide a very unique set of markers for separating this source from other biomass sources. Additionally, resin acids from observed in the emissions from spruce and pine emissions provide unique species associated with these emissions (and observed previously).*

*At the present moment, to provide insight into the usefulness of these markers within the context of ambient measurements, or against source apportionment methods, we would require a robust dataset of comparable data to test these markers and average emission profiles against. "*

**Minor comments**

The abstract part is not transparent and brief. See line 17 – 20, I recommend cutting this sentence into two separate sections.

Response: The original sentence has been rewritten into two sentences as shown below on line 17 to 20.

*"In this study, we systematically estimated the emission factors and characterized the primary OA (POA) chemical composition with the AMS and the extractive electrospray ionization time-of-flight mass spectrometer (EESI-TOF) for the first time. The study was conducted on a variety of solid fuels, including beech logs, spruce and pine logs, spruce and pine branches and needles, straw, cow dung, and plastic bags."*

Line 42:45: "… Southeast Asia… developing regions,…India", Please rewrite these sentences, as these regions are involved with each other.

Response: They are rephrased as below on line 43 to 45.

*"In Southeast Asia, haze events are mainly attributed to the wildfires, agricultural waste burning, and peatland fires (Adam et al., 2021). In India, more than half of households use inefficient stoves for cooking, burning solid fuels such as firewood, charcoal, crop residues, and cow dung (Census of India, 2011)."*

Line 117: high-resolution time-of-flight AMS (HTOF-AMS)? OR H*R*-TOF?

Response: We think both abbreviations are fine as long as it is clear in the manuscript. We prefer to refer it as HTOF-AMS, as it is consistent with the abbreviation form of the name from its producer Aerodyne Research, Inc., which can be found here: https://www.aerodyne.com/product/aerosol-mass-spectrometer.

Line 171, why is $p<0.1$ chosen?

Response: The significance level α typically is 0.1, 0.05, and 0.01. When we choose α = 0.1 to formulate the hypotheses, it means there is 10% chance of rejecting the null hypothesis when the null hypothesis is true (Type I error). In the two-tailed test we used, for each tail (significantly high or significantly low) the α is equivalent to 0.05 in the one-tailed test as the cutoff. This is a practically useful cutoff. When α is set to 0.05, the criteria is too strict to effectively select markers for straw, while, an α higher than 0.1 runs the risk of committing the Type I error.

Line 692, static p?

Response: changed.

**Reference**

Engling, G., Carrico, C. M., Kreidenweis, S. M., Collett Jr, J. L., Day, D. E., Malm, W. C., Lincoln, E., Min Hao, W., Iinuma, Y., and Herrmann, H.: Determination of levoglucosan in biomass combustion aerosol by high-performance anion-exchange chromatography with pulsed amperometric detection, Atmos. Environ., 40, 299-311, https://www.sciencedirect.com/science/article/pii/S1352231006005802, 2006.

Fabbri, D., Modelli, S., Torri, C., Cemin, A., Ragazzi, M., and Scaramuzza, P.: GC-MS determination of levoglucosan in atmospheric particulate matter collected over different filter materials, J Environ Monit, 10, 1519-1523, https://www.ncbi.nlm.nih.gov/pubmed/19037493, 2008.

Pashynska, V., Vermeylen, R., Vas, G., Maenhaut, W., and Claeys, M.: Development of a gas chromatographic/ion trap mass spectrometric method for the determination of levoglucosan and saccharidic compounds in atmospheric aerosols. Application to urban aerosols, J Mass Spectrom, 37, 1249-1257, https://www.ncbi.nlm.nih.gov/pubmed/12489085, 2002.

---

## Author Comment (AC3)

Reviewer #3

The study utilized two methods, namely the aerosol mass spectrometer (AMS) and the extractive electrospray ionization time-of-flight mass spectrometer (EESI-TOF), to investigate the molecular composition of primary organic aerosol derived from various solid waste sources. The study also reported emission factors and presented spectra obtained from the AMS and EESI-TOF instruments. While the article is well-written, there are some areas that require improvement before publication.

We thank the referee for the valuable comments which have greatly helped us improve the manuscript. We will have the reviewer comments in black, address the comments in blue, and modified sentences in blue and italics.

1. The title and abstract/keywords suggest a desire for insights into the molecular-level composition and markers. Although Section 3.4 briefly discusses individual markers, the discussion seems disconnected from the figures and tables and is not adequately reflected in the abstract and conclusion. Enhancing the presentation and discussion of these markers would make the work more captivating.

   Response: To enhance the presentation and discussion of the markers for the use of source separation as we aim for, a figure as Figure S6 in the SI and the corresponding discussion on line 394 to line 404 in the main text.

   "

[Figure]

   *Figure S6. The scatter plots of marker ions from (A) AMS and (B) EESI-TOF.*

   *From the perspective of source-apportionment, ions that are primarily associated with a specific emission source and exhibit minimal contribution from other sources can be considered as potent in use. To show the ability of these markers for source separation, the contribution of two markers for the same source from Table S2 and Table S3 that possess small p-value with high FC are plotted among studied fuels. As shown in Figure S6, these markers measured by the AMS have relatively higher contribution in one specific fuel, which makes the fuel distinctive from others. Nonetheless, one would need to coordinate with more tracers to draw a conclusive diagnosis because the presence of these markers in other fuels. Given this scenario, the markers that have significantly low contribution (FC < 1) in a specific fuel could shed the lights on. In contrast, markers observed from the EESI-TOF is more practical for utilization as most of them are unique. As the markers listed in Table S2 and Table S3 are many, we could not thoroughly discuss here. However, due to the limited fuel types, referencing more markers can provide more confidence in source identification."*

2. Most of the figures predominantly focus on the overall composition in a statistical manner. It would be helpful if the authors elaborate on whether they consider the composition or individual molecules more important for source apportionment. This clarification would enrich the paper.

Response: The regarding concern has been elaborated in our conclusion section as follows on line 448 to line 458.

*"Future studies will probe the usefulness of these markers, if they are long lived enough in the atmosphere to provide useful separation between complex emission sources shown here. This will either focus on online measurements in polluted regions or from offline filter analysis from similar regions. Clearly, the dominant biomass burning markers (levoglucosan and others) are not robust to be used to separate different biomass sources, though they are robust for identification of general biomass burning aerosol. Nitrogen containing compounds emitted from cow dung emissions can provide a very unique set of markers for separating this source from other biomass sources. Additionally, resin acids from observed in the emissions from spruce and pine emissions provide unique species associated with these emissions (and observed previously).*

*At the present moment, to provide insight into the usefulness of these markers within the context of ambient measurements, or against source apportionment methods, we would require a robust dataset of comparable data to test these markers and average emission profiles against."*

3.  Although the introduction mentions the possibility of using Positive Matrix Factorization (PMF) for data analysis, it was not employed in this study. Are there any plans to utilize PMF for analyzing this dataset? This aspect could be addressed to provide a clearer understanding.

    Response: The PMF mentioned in the introduction is an important approach to utilize the result of this study in the ambient measurement, as it uses characteristic ions from mass spectra to diagnose the sources of ambient aerosol. For the single-source static data studied in this paper, PMF is not applicable. The following sentence has been added in the conclusion to clarify this on line 456 to line 458.

    *"At the present moment, to provide insight into the usefulness of these markers within the context of ambient measurements, or against source apportionment methods, we would require a robust dataset of comparable data to test these markers and average emission profiles against."*

4.  On page 6, line 204, the reasoning for excluding the mass spectrum of plastic bags burning is not convincing. The statement suggests that the observed CxHyOz family in the spectrum (23%) is more likely due to emissions remaining in the tubing from other fuels rather than from the plastic bags themselves, given that polyethylene is their main component. It would be valuable to include the results of plastic bag burning in the main figures and provide further explanation regarding the expected products from burning plastic bags.

    Response: The mass spectrum of plastic bags burning has been included in Figure 1 after correction. The correction has been explained in the supplement as follows (Line 11 to 23).

    *"**Plastic bags burning emission correction.** In three out of four plastic bags burning experiments, the mass spectrum at the middle to end burning stages had considerable $C_xH_yO_z$ family contribution (~23%). It is unlikely from the plastic bags, given the fact that polyethylene is the main component of plastic bags, but from the emission of other fuels remaining in the chimney. As the combustion progressed, the chimney was heated, and the volatile substances remaining on the chimney evaporated and were then partitioned to the particles for detection. However, at the early stage, before the chimney got hot, the mass spectra consisted mainly of hydrocarbons (see Figure S2a). Therefore, we take only the early burning stage of these three burning experiments into account for the average mass spectrum in Figure 1(f). The absolute concentration of the three AMS mass spectra derived from the early-stage burning is scaled to the uncontaminated burning experiment ions based on m/z 81 and m/z 83, which are stable and characteristic for hydrocarbons. The difference on average is 0.4% ± 1.0% which is very minor as shown in Figure S2b. The mass spectra of three contaminated burning over the whole burning stages indicate that the measured organics was 14.6% ± 8.7% overestimated. Correspondingly, the emission factors for PM and OM are corrected for each plastic bags burning.*

[Figure]

*Figure S2 (a) The time series of some ions measured by the AMS during the plastic bags burning for the contaminated case; (b) the mass spectrum comparison of uncontaminated plastic bags burning experiment at the top v.s. the average of 3 early-stage burning at the bottom."*

5.  On page 11, line 402, the statement "the markers identified by the AMS have greater replicability and by EESI-TOF are more distinctive, thus providing an important reference for the source apportionment" lacks clarity. Please rephrase this sentence to enhance its meaning.

    Response: It is explained in Lines 327-331 in Section 3.4 in the original manuscript. It has been rephrased as below on line 438 to line 440.

    "*The markers identified by the AMS have greater replicability indicated by the smaller p-value, and by EESI-TOF are more distinctive which can be seen from the higher FC. Therefore, the employment of both provides an important reference for the source apportionment.*"

---

## Author Comment (AC4)

Reviewer #4

The article by Jun Zhang et al. describes the emissions of biomass burning using different devices both on molecular level and on bulk. The emission factors and composition of emissions are shown to depend on the burned fuel and some marker compounds are identified. Article is well written and easy to read and follow. The information provided in the article is important for scientific community and the in my opinion the article should be accepted after revision. I have few comments mainly intending to clarify some aspects of the manuscript:

We would like to thank the reviewer for the comments and suggestions to improve the current work. We will have the reviewer comments in black, address the comments in blue, and modified sentences in blue and italics.

1. The results section is referring to PM. Please give the corresponding size class, PM1? PM0.5? is it always always same, from the SMPS, not AMS? I think it is fine if you add a sentence about this to experimental and then continue using PM abbreviation.

    Response: The SMPS applied in this work scans particle diameter from 16 to 638 nm, and the AMS uses PM1 aerodynamic lens as introduced in the method section. Particle size distribution measured by the SMPS does not appear to extend past 638 nm. Therefore, the PM referred to in the result section is in the range of PM1.

    It has been clarified in the method section below on line 129.

    "*Therefore, the class of the PM in this study belongs to PM1*"

2. Please describe the holding tank and its role in more detail. Why it was used and what was the impact on PM?

    Response: The holding tank and its role are added as follows (Line 111 to line 113).

    "*The holding tank is a stainless steel container (1 m$^3$) used to store emissions. It is also designed for averaging the emissions at different combustion efficiency in order to fully characterize the emission in the real ambient.*"

3. Fig S1. maybe replace Dekati with Dekati Ejector

    Response: The "Dekati" has been replaced with "*Dekati Ejector*" in Figure S1.

4. Line 204: "The mass spectrum of plastic bags burning is not shown because considerable CxHyOz family was observed (23%), but it is more likely from the emission remaining in the tubing of other fuels than from the plastic bags given the fact that polyethylene is the main component of it." was this only affecting AMS results or are all plastic bag burning results suffering from this? did you measure blank values to ensure this same phenomenon was not affecting other results also?

    Response: The AMS results including mass spectrum, organic mass concentration, and elemental ratio, as well as the corresponding emission factors, have been corrected. The contamination of analysis of plastic bags has been added in the supplement as below. The overestimation of POA from plastic bags burning is 14.6% on average, which is mainly from semi-volatile oxygen-containing species. These species are important components emitted by other biomass materials used in this study. Therefore, the overestimation is minor for other burning fuels. The following discussion has been added from line 11 to line 23 in the SI.

    "*Plastic bags burning emission correction. In three out of four plastic bags burning experiments, the mass spectrum at the middle to end burning stages had considerable C$_x$H$_y$O$_z$ family contribution (~23%). It is unlikely from the plastic bags, given the fact that polyethylene is the main component of plastic bags, but from the emission of other fuels remaining in the chimney. As the combustion progressed, the chimney was heated, and the volatile substances remaining on the chimney evaporated and were then partitioned to the particles for detection. However, at the early stage, before the chimney got hot, the mass spectra consisted mainly of hydrocarbons (see Figure S2a). Therefore, we take only the early burning stage of these three burning experiments into account for the average mass spectrum in Figure 1(f). The absolute concentration of the three AMS mass spectra derived from the early-stage burning is scaled to the uncontaminated burning experiment ions based on m/z 81 and m/z 83, which are stable and characteristic for hydrocarbons. The difference on average is 0.4% ± 1.0% which is very minor as shown in Figure S2b. The mass spectra of three contaminated burning over the whole burning stages indicate that the measured organics was 14.6% ± 8.7% overestimated. Correspondingly, the emission factors for PM and OM are corrected for each plastic bags burning.*"

[Figure]

*Figure S2 (a) The time series of some ions measured by the AMS during the plastic bags burning for the contaminated case; (b) the mass spectrum comparison of uncontaminated plastic bags burning experiment at the top v.s. the average of 3 early-stage burning at the bottom."*

5.  Line 199: how the PM/OM/BC Emission factors compare with literature values? are the literature available for e.g. plastic bags?

    Response: The emission factors for PM, OM, and BC are compared to the literature as follows on line 210 to 214.

    *"In general, the EF of PM, OM, and BC agrees well with some previous literature (Fang et al., 2017; Goetz et al., 2018; Tissari et al., 2008). Nevertheless, the reported EF values are highly dependent on the burning method (e.g. stove type) and combustion efficiency (Bertrand et al., 2017). Additionally, the reported EFs for plastics vary substantially with their composition, and the EF of the pure PE plastic bags are not often reported (Jayarathne et al., 2018; Wu et al., 2021; Hoffer et al., 2020)."*

6.  Are the terms f44/f60/fxx etc explained somewhere? I may have missed this

    Response: Yes, they are explained in Line 212-Line 213 in the original manuscript, but this was extended to include the meaning further (Line 224-Line 226):

    *"Among these ions, the mass fractions of m/z 44 ($f_{44}$, mostly $CO_2^+$), mass fraction of m/z 43 ($f_{43}$, mostly $C_2H_3O^+$ and $C_3H_7^+$), and mass fraction of m/z 60 ($f_{60}$, mostly $C_2H_4O_2^+$) have the largest impact on the oxidation state of the aerosol."*

7.  Authors are using terms OA, OM, OC, and POA for the organic fraction. Maybe you could clarify what is the difference between them.

    Response: OA includes POA, and equivalent to OM. OC only refers to the carbon in organics. The ratio of OM and OC are used in the emission factor calculation. In the result section, we use POA throughout denoting the organic fraction in the measurement. These concepts have been clarified in the manuscript as follows

    *"Emissions from combustion are a major source of primary organic aerosol (POA),..."* (Line 36)

    *"Organic aerosol (OA, including POA and SOA) source apportionment has been widely studied using receptor models,..."* (Line 55)

    *"where $\Delta CO$, $\Delta CO_2$, $\Delta THC$, $\Delta OC$, and $\Delta BC$ are the background-corrected carbon mass concentrations of CO, $CO_2$, THC, OC (organic carbon), and BC."* (Line 157-158)

    *"OC was calculated from the ratio of organic aerosol and the ratio of organic mass (OM) to OC (OM/OC) measured by AMS"* (Line 158-159)